# Comparative Transcriptomics Analysis Reveals the Differences in Transcription between Resistant and Susceptible Pepper (*Capsicum annuum* L.) Varieties in Response to Anthracnose

**DOI:** 10.3390/plants13040527

**Published:** 2024-02-15

**Authors:** Yixin Wang, Bin Chen, Chunyuan Cheng, Bingkun Fu, Meixia Qi, Heshan Du, Sansheng Geng, Xiaofen Zhang

**Affiliations:** 1Beijing Vegetable Research Center, Beijing Academy of Agriculture and Forestry Sciences, Beijing 100097, China; wangyixin@nercv.org (Y.W.); chenbin@nercv.org (B.C.); 15938218023@163.com (C.C.); duheshan@nercv.org (H.D.);gengsansheng@nercv.org (S.G.); 2College of Horticultural, China Agricultural University, Beijing 100097, China; saber1127235370@163.com (B.F.); qimeixia0702@163.com (M.Q.)

**Keywords:** pepper, anthracnose, GO enrichment, *Colletotrichum capsici*, transcription

## Abstract

Pepper (*Capsicum annuum* L.) is a herbaceous plant species in the family *Solanaceae*. Capsicum anthracnose is caused by the genus *Colletotrichum.* spp., which decreases pepper production by about 50% each year due to anthracnose. In this study, we evaluated the resistance of red ripe fruits from 17 pepper varieties against anthracnose fungus *Colletotrichum capsici*. We assessed the size of the lesion diameter and conducted significance analysis to identify the resistant variety of B158 and susceptible variety of B161. We selected a resistant cultivar B158 and a susceptible cultivar B161 of pepper and used a transcription to investigate the molecular mechanisms underlying the plant’s resistance to *C. capsici*, of which little is known. The inoculated fruit from these two varieties were used for the comparative transcription analysis, which revealed the anthracnose-induced differential transcription in the resistant and susceptible pepper samples. In the environment of an anthrax infection, we found that there were more differentially expressed genes in resistant varieties compared to susceptible varieties. Moreover, the response to stimulus and stress ability was stronger in the KANG. The transcription analysis revealed the activation of plant hormone signaling pathways, phenylpropanoid synthesis, and metabolic processes in the defense response of peppers against anthracnose. In addition, ARR-B, AP2-EREBP, bHLH, WRKY, and NAC are associated with disease resistance to anthracnose. Notably, WRKY and NAC were found to have a potentially positive regulatory role in the defense response against anthracnose. These findings contribute to a more comprehensive understanding of the resistance mechanisms of red pepper fruit to anthracnose infection, providing valuable molecular insights for further research on the resistance mechanisms and genetic regulations during this developmental stage of pepper.

## 1. Introduction

Anthracnose refers to widespread plant diseases caused by the *Colletotrichum* species. These fungi can severely damage various vegetables, flowers, crops, and fruit trees (e.g., plant blight, fruit decay, and leaf spot disease), resulting in serious economic losses [1,2]. Pepper (*Capsicum annuum* L.) is a popular solanaceous vegetable that is rich in ascorbic acid, zeaxanthin, capsaicin, lutein, and β-carotene, which has beneficial effects on humans [3]. It can be processed into food products, including chili sauce, chili oil, and other condiments. In addition to its edible value, pepper also has an economic value. C apsanthin and capsaicin are used in industrial and medical fields. Thus, pepper is an important cash crop cultivated globally [4]. Anthracnose of pepper, which was first detected in India [5], is mainly caused by the following four fungi: *Colletotrichum gloeosporioides*, *Colletotrichum acutatum*, *Colletotrichum capsici*, and *Colletotrichum coccodes* [6,7,8,9,10,11,12]. Anthracnose decreases annual pepper production by 29.5% in India, 20–80% in Vietnam, and 10% in South Korea, leading to considerable economic losses [13].

Anthracnose is a common and significant fungal disease that occurs in peppers, also known as ring spot disease or circular spot disease. It mainly affects mature fruits and leaves. The leaf lesions initially appear as pale green water-soaked spots, which gradually turn brown and become nearly circular. After the fruits are infected, they develop brown circular or irregular concentric ring lesions, which have black or red small dots on the surface. When it is moist, there is a discharge of a light red sticky substance. Pathogen infections trigger the synthesis of defense-related compounds, including lignin, hydroxyproline, phenolic substances, and various enzymes [14]. However, the resistance mechanism of pepper to anthracnose is not clear. Therefore, understanding the molecular basis of *C. capsici* pathogenicity and the mechanisms that confer resistance to *C. capsici* is crucial. Additionally, it is essential to investigate how a *C. capsici* infection affects metabolism and signal transduction pathways.

In a recent study, C_2_H_2_ zinc finger transcription factor (TF) genes were identified, and their expression levels in diseased pepper were analyzed to further clarify how the pepper response to anthracnose is regulated [15]. Using CRISPR/Cas9 technology to knock out *CaERF28* can significantly increase the resistance of pepper to *Colletotrichum truncatum* [16]. In addition, an earlier transcriptome sequencing (RNA-seq) analysis of resistant and susceptible varieties detected 331 differentially expressed genes (DEGs), which are mainly related to metabolism, defense responses, and hormone regulation [17].

High-throughput RNA-seq analyses are useful for detecting comprehensive gene expression changes and determining the responses of gene regulatory networks to stress factors in plants [18]. For example, an RNA-seq-based comparison of the responses of wild rice and cultivated rice roots indicated the resistance of wild rice to *Magnaporthe oryzae* [19], which is related to lipid metabolism, WRKY TFs, jasmonic acid, ethylene, lignin, and phenylpropanoid and diterpenoid metabolism. Yu et al. [20] examined the gene expression of two gray spot-resistant maize varieties at several time-points post infection; they identified the pathways involved in the maize response to the pathogen causing gray spots (e.g., salicylic acid response, protein phosphorylation, redox process, and carotenoid biosynthesis). Szymanski et al. [21] combined a transcriptome analysis of tomato lines with pathogen sensitivity assays to identify the genes related to defense responses and fungal resistance.

Breeders have used traditional and modern molecular breeding techniques to generate anthracnose-resistant pepper varieties containing resistance genes. However, the molecular resistance mechanism of pepper to anthracnose remains unclear. This experiment selected pepper cultivar resistant and susceptible to *Colletotrichum capsici* (TJ3-3), conducted RNA-seq analysis, and functionally analyzed differentially expressed genes to explore important genes and metabolic pathways related to disease resistance. This study may provide a foundation for clarifying the molecular mechanism underlying the resistance of pepper to anthracnose and for establishing an efficient method for increasing anthracnose resistance in pepper via molecular breeding.

## 2. Results

### 2.1. Screening of Pepper Materials for Disease Resistance

By analyzing the lesion diameter of pepper varieties, it was found that B158 had the smallest average lesion diameter and B161 had the largest average lesion diameter (Appendix A). Meanwhile, there is a highly significant difference between B158 and B161 at the *p* < 0.05 level. Therefore, B158 is a disease-resistant variety and B161 is a disease-susceptible variety (Figure 1 and Figure 2).

### 2.2. RNA-Seq and de Novo Transcription Assembly

The B158-CK and B161-CK samples were mock inoculated with sterile water, whereas the B158-T and B161-T samples were inoculated with the pathogen (*C. capsici*). Twelve RNA-seq libraries were constructed and sequenced, resulting in more than 55,746,766 raw reads per library. An average of 83.4 million clean reads were obtained after filtering for quality (GC content ≥ 42% and Q30 ≥ 94%) (Table 1).

### 2.3. DEGs Analysis of Pepper in Response to C. capsici Infection

A comparison between B161-CK and B161-T detected a total of 1306 differentially expressed genes (DEGs), with 1129 genes being up-regulated and 177 genes being down-regulated. Similarly, in the B158-CK vs. B158-T comparison, a total of 5384 DEGs were identified, comprising 2887 up-regulated genes and 2497 down-regulated genes. (Figure 3A–C). Moreover, a total of 329 DEGs were found to be common to both the B158-CK vs. B158-T and B161-CK vs. B161-T comparisons. In contrast, here were 5055 DEGs exclusively identified in the B158-CK vs. B158-T comparison and 977 DEGs exclusively identified in the B161-CK vs. B161-T comparison. (Figure 3D). The results indicate that a significant number of genes are involved in the response to anthracnose in resistant cultivars.

### 2.4. Functional Characterization according to GO Terms and KEGG Pathways

The gene expression level was estimated using the gene expression abundance corresponding to the FDR value, and a differential gene analysis was performed with *p* ≤ 0.05 and |log_2_Fold Change| ≥ 1 as the screening criteria. The sequencing results showed 6361 DEGs in the resistance cultivar and susceptible cultivar during the infection process. A pairwise comparison of DEGs was performed between the B158 and B161 at specific time points, and 5384 and 1306 DEGs were identified at 24 and 48 h. We mapped DEGs to gene ontology terms to find significantly enriched GO entries. The GO analysis revealed that DEGs were classified into three ontologies, including biological processes (BPs), molecular functions (MFs), and cellular components (CCs).

We conducted GO and KEGG enrichment analyses on the differentially expressed genes (DEGs) in these two varieties. We further annotated these DEGs with 53 enriched GO terms for variety B158 and 49 enriched GO terms for variety B161. Most DEGs (*p* ≤ 0.05) enriched in BPs were mainly enriched in metabolic processes, cellular processes, single-organism processes, and in response to stimulus. MFs were mainly enriched in catalytic activity, binding, and transporter activity, while CCs were mainly enriched in cells, cell parts, and organelles (Appendix A).

There were 133 significantly enriched biological process GO terms assigned to the DEGs identified by the B158-CK vs. B158-T comparison. The top three terms were response to stimulus (1174 DEGs), response to stress (747 DEGs), and phenylpropanoid biosynthetic process (64 DEGs) (Figure 4A). The DEGs detected by the B161-CK vs. B161-T comparison were annotated with 167 significantly enriched biological process GO terms, of which rRNA metabolic process (51 DEGs), plastid organization (53 DEGs), and ncRNA metabolic process (57 DEGs) were the top three terms (Figure 4B). The 927 and 252 DEGs identified by the B158-CK vs. B158-T and B161-CK vs. B161-T comparisons were associated with 130 and 103 KEGG pathways, respectively (Figure 5). The three significantly enriched KEGG pathways (Q < 0.05) among the DEGs revealed by the B161-CK vs. B161-T comparison were carotenoid biosynthesis, carbon fixation in photosynthetic organisms, and carbon metabolism (Figure 5B). For the B158-CK vs. B158-T comparison, there were 17 significantly enriched KEGG pathways (Q < 0.05) among the DEGs, of which metabolic pathways, biosynthesis of secondary metabolites, and phenylpropanoid biosynthesis were the top three (Figure 5A).

### 2.5. Analysis of the DEGs Exclusive to B158

The 5055 DEGs detected only by the B158-CK vs. B158-T comparison may be useful for eliminating the differences in the genetic background of the plant varieties and may reveal important insights into the increased resistance of B158 to anthracnose (Figure 1D). We mapped 5055 DEGs to gene ontology terms to find significantly enriched GO entries. The biological processes related to disease resistance include differentially expressed genes (DEGs) involved in the phenylpropanoid biosynthesis pathway, phenylpropanoid metabolism, responses to stress, and defense against fungi. Molecular functions enriched in oxidoreductase activity, and cellular components enriched in the cell wall pathway. 5055 DEGs were enriched to KEGG, and the enrichment pathways related to disease resistance were identified. DEGs was enriched in the biosynthesis of secondary metabolites, phenylpropanoid biosynthesis, plant hormone signal transduction, glutathione metabolism and amino sugar and nucleotide sugar metabolism.

Phenylpropanoid-related genes were among the 20 terms with the highest enrichment degree in the GO and KEGG analysis (Figure 6). Phenylpropanoid-related GO terms were assigned to 61 up-regulated genes and 5 down-regulated genes. Moreover, phenylpropanoid biosynthesis was identified as an enriched KEGG pathway among 59 up-regulated genes and 6 down-regulated genes. Additionally, the response to the fungus GO term, which is also associated with anthracnose resistance, was assigned to 78 up-regulated genes and 20 down-regulated genes. Moreover, we observed an up-regulation of genes encoding 4CL (CA.PGAv.1.6.scaffold546.13) and CCoAOMT (CCoAOMTCA.PGAv.1.6.scaffold354.27). Plant hormone signal transduction, another factor influencing anthracnose resistance, was identified as an enriched KEGG pathway among 37 up-regulated genes and 29 down-regulated genes. It is fascinating to observe the significant up-regulation of genes involved in the plant hormone signaling pathway, namely, auxin (LAX3, CA.PGAv.1.6.scaffold393.6), ethylene (ERF1B, CA.PGAv.1.6.scaffold112.1), and the pathogens related (PR1B1, CA.PGAv.1.6.scaffold504.17) (Appendix A).

### 2.6. Analysis of the TFs

A total of 5055 differentially expressed genes (DEGs), including 219 transcription factor (TF) genes, were identified. Among the TF genes, several families were found to be associated with disease resistance. These families include ARR-B (35 genes), AP2-EREBP (22 genes), bHLH (18 genes), WRKY (17 genes), and NAC (14 genes).

## 3. Discussion

### 3.1. Response of the Resistant Variety B158 to Anthracnose

The transcription analysis identified 5055 genes that were expressed only between B158-CK and B158-T (i.e., the resistant variety). There were only a few up-regulated genes revealed by the B161-CK vs. B161-T comparison. According to the B158-CK vs. B158-T comparison, the number of genes up-regulated in DEGs significantly increased after the inoculation of *C. capsici*, these up-regulated genes including those encoding negative regulators and positive regulators of immune responses, which genes encode proteins associated with antioxidant activity.

The results of the GO and KEGG analyses indicated the DEGs were mainly involved in phenylpropanoid biosynthesis, secondary metabolite biosynthesis, ribonucleic acid metabolism, and cell signaling. The phenylpropanoid biosynthesis pathway is one of the three main plant secondary metabolic pathways, which lead to the formation of diverse phenylalanine metabolisms, including flavonoids, lignin, anthocyanins, and terpenoids (Figure 7). Phenylpropane compounds are important for plant growth and development, and they also contribute to stress responses, thereby enhancing disease resistance [22,23]. The above-mentioned results suggest that the B158 response to anthracnose involves the induced expression of many defense response-related genes.

### 3.2. Involvement of TFs in Biological Stress Networks

Transcription factors have a crucial impacts on plant growth and development. In this study, we found that ARR-B, AP2-EREBP, bHLH, WRKY, and NAC are all important regulators of defense responses in resistance cultivar B158.

Many studies have confirmed that WRKY and NAC TFs influence plant disease resistance. For example, Yu et al. [24] reported that pathogen infections and the application of exogenous salicylic acid (SA), methyl jasmonate, and methyl viologen can induce *GhWRKY15* expression in cotton seedlings. The overexpression of *GhWRKY15* significantly increases the resistance to *Colletotrichum gossypii* and *Phytophthora parasitica.* A recent study clarified the effects of WRKY TFs on the response of *Lilium* species to *Botrytis* fungal diseases [25]. The expression of CaWRKY27 was up-regulated by SA, MeJA and ET, the overexpression of CaWRKY27 in tobacco increased resistance to *R. solanacear*, and the expression of genes related to the course of the disease and SA, JA and ET genes were up-regulated, indicating that CaWRKY27 played a positive regulatory role in disease resistance defense through SA, JA and ET-mediated signaling pathways [26]. In our study, we found that transcription factors WRKY40 (CA.PGAv.1.6.scaffold407.64), WRKY22 (CA.PGAv.1.6.scaffold532.44), and WRKY28 (CA.PGAv.1.6.scaffold585.27) were significantly up-regulated. Previous studies have shown that these three transcription factors positively regulate the viral response which is consistent with the results of this study (Figure 8B).

The overexpression of grape VvNAC1 in Arabidopsis enhances resistance to gray mold and downy mildew by regulating defense genes [27]. The overexpression of VaWRKY10 in Arabidopsis and seedless grapes enhances resistance to gray mold. the overexpression of OsbHLH057 enhances rice’s disease and drought resistance [28]. In rice, *ONAC066* encodes a protein that positively regulates rice resistance to blast and bacterial blight by modulating the abscisic acid (ABA) signaling pathway as well as the accumulation of sugars and amino acids [29]. The orphan protein TaFROG interacts with TaNACL-D1, which is an NAC TF that participates in the response of *Triticeae* species to disease [30]. CaNAC2c enhances immunity to RSI by activating jasmonic acid-mediated immunity and H_2_O_2_ accumulation [31]. In resistant varieties, we found significant expression of CA.PGAv.1.6.scaffold868.1 NAC transcription factors, indicating that NAC may positively regulate resistance to *C. capsici* (Figure 8A).

### 3.3. Analysis of Disease Resistance-Related Pathways

Plants have evolved various multi-branched phenylalanine metabolism pathways that produce many metabolites, including flavonoids, lignin, lignans, and cassia bark acid amides. In a recent study on cherry, the proteins encoded by *CAD*, *POD*, *CCoAOMT*, *4CL*, *CCR*, and *COMT* genes were revealed to affect the lignin content, suggestive of their possible role in the resistance to *Alternaria alternata* [32]. Lignin can contribute to the scavenging of reactive oxygen species. Moreover, the free radicals produced during lignin metabolism may inactivate fungal cells. In Arabidopsis, *CAD-C* and *CAD-D*, which are major genes involved in lignin biosynthesis, encode important components of the defense responses to virulent and avirulent strains of *Pseudomonas syringae* pv. *tomato* mediated by the salicylate defense pathway [33]. In cotton, GhROP6 helps regulate the expression of the lignin synthesis-related genes *GhCCR-1*, *GhF5H-1*, *GhCCoAOMT-2*, and *GhCCoAOMT-3*, leading to increased resistance to Verticillium wilt [34]. In this study, the DEGs associated with phenylpropanoid biosynthesis were mainly involved in lignin synthesis, resulting in the production of p-hydroxy-phenyl lignin, guaiacyl lignin, g-hydroxy-guaiacyl lignin, and syringyl lignin. Accordingly, lignin is an important branch of the phenylpropanoid biosynthesis pathway. We observed that the expression levels of *4CL* (CA.PGAv.1.6.scaffold546.13) and *CCoAOMT* (CCoAOMTCA.PGAv.1.6.scaffold354.27) were up-regulated in KANG (Figure 9 and Appendix A), which modulated the lignin content, implying they may influence the resistance of pepper to *C. capsici*. In the chickpea root, microRNA397 reportedly regulates the tolerance to drought and fungal infections by controlling lignin deposition [35]. Phytohormones play a critical role in plant responses to viral infections [36]. In the current study, we detected the up-regulated expression of ABA and SA signal transduction pathway genes. The transduction of the ABA signal is initiated when ABA enters cells via the membrane or cytosolic receptors (e.g., PYR/PYL/RCAR), which leads to the activation of the downstream PP2C phosphokinase and SnRK2 through the phosphorylation cascade. The subsequent activation of downstream TFs induces the expression of downstream genes, ultimately affecting seed germination and adaptations to stress. In this study, we observed that *PYL2* (CA.PGAv.1.6.scaffold370.81) and *PYL4* (CA.PGAv.1.6.scaffold243.38) expression levels were up-regulated by anthracnose (Appendix A). Therefore, the ABA receptor PYL along with the regulatory factors PP2C and their downstream components form a complete ABA signaling pathway during the KANG response to anthracnose.

The WRKY TFs, which are named because of their conserved WRKYGQK sequence, are critical components of the signaling network controlling plant responses to pathogens [37]. Previous studies have shown that many WRKY TFs are involved in SA-mediated defense signaling pathways. In Arabidopsis, *WRKY38* and *WRKY62* expression is induced by SA and modulates the SA signaling pathway [3]. In grapevine, VqWRKY31 enhances the resistance to powdery mildew [38] through its effects on SA-related defense signaling. In apple, MdWRKY17 promotes the expression of *MdDMR6* (associated with SA degradation), leading to increased susceptibility to *Colletotrichum fructicola* [39]. In citrus, exogenous SA regulates *CsWRKY70* expression and methyl salicylate synthesis to enhance the resistance to *Penicillium digitatum* [40]. In tobacco, NbWRKY40 regulates the expression of SA-related genes in response to tomato mosaic virus [41].

Previous research has demonstrated that WRKY TFs play a key role in the pepper defense response to pathogens through the SA signaling pathway. The expression of *CaWRKY40* alters various biological processes and is crucial for the response to an infection by *Ralstonia solanacearum* [42,43,44]. Similarly, the expression of *CaWRKY27* [26], *CaWRKY6* [45], *CaWRKY22* [46], *CaWRKY41* [47], *CaWRKY30* [48], *CaWRKY28* [49], and *CaWRKY27b* [43] positively regulates resistance to an *R. solanacearum* infection; however, the expression of *CaWRKY58* [50] and *CaWRKY40b* [51] has the opposite effect. The CaWRKY50 TF negatively regulates the resistance of pepper to *Colletotrichum scovillei* through its effects on SA-mediated signaling and the antioxidant defense system [52]. The WRKY TFs can bind specifically to W-box cis-elements (TTGACC/T) in promoters to regulate the expression of defense-related genes. The WRKY70 TF increases the expression of SA signaling-related genes (*TGA6* and *TGA70*), which influences the susceptibility to *Verticillium dahliae* toxins [50]. In the current study, 17 genes encoding WRKY TFs (e.g., WRKY22, WRKY28, WRKY40, and WRKY70) were among the DEGs identified in B158, which was resistant to *C. capsici*. Furthermore, the expression levels of SA signaling pathway-related genes were up-regulated. Considered together, these findings suggest that the resistance of B158 to anthracnose is mediated by WRKY TFs that control SA signaling-related gene expression.

## 4. Materials and Methods

### 4.1. Colletotrichum capsici Culture

The anthracnose pathogen (*Colletotrichum capsici* TJ-3-3) used in this study was provided by Xiulan Xu from the Vegetable Research Center of the Beijing Academy of Agricultural and Forestry Sciences. The pathogen was transferred to potato dextrose agar (Merck, Johannesburg, South Africa) and then incubated at 25 °C for 7–10 days. The colonies were used to prepare the anthracnose inoculum.

Plant varieties and the inoculation a total of 17 pepper (*Capsicum annuum* L.) varieties (B2, B19, B41, B68, B70, B71, B72, B73, B74, B75, B76, B77, B78, B85, B87, B158, and B161) were obtained from the Vegetable Research Center of the Beijing Academy of Agricultural and Forestry Sciences. The plant varieties were bred and grown in a multi-span greenhouse at the research center.

Unblemished ripe (50 days after anthesis) pepper fruits with a uniform size were surface-disinfected with 75% ethanol. Each fruit was wounded using a syringe needle, after which the conidial suspension (3 × 10^6^ spores/mL) was added to the wound site (6 μL per site). The control fruit were similarly wounded but were mock inoculated with sterilized distilled water. Each treatment was completed using three replicates, each comprising nine fruits. The inoculated fruit were placed in a bread box containing a moistened paper towel to maintain humidity during the incubation at 26 °C. Samples of the infection site were collected at 7 days post inoculation, immediately frozen in liquid nitrogen, and stored at −80 °C.

### 4.2. Statistical Analysis of Pepper Disease Spot Data

On the seventh day of inoculation with pepper anthracnose, we measured the diameter of the lesion. We imported the measured lesion diameter into Microsoft Excel 2021 for preliminary organization and used the software Graphpad 9 for the significance analysis. *p* < 0.05 indicated significant difference, and *p* < 0.01 indicated an extremely significant difference.

### 4.3. RNA Extraction and Sequencing Library Construction

The most resistant and susceptible varieties were used for the transcriptome analysis. At 7 days post inoculation, the pericarp tissue from the pepper fruit lesions was collected and immediately frozen in liquid nitrogen for subsequent RNA extraction. For the treatment group containing the B158 resistant variety (B158-T) and the B161 susceptible variety (B161-T) inoculated with *C. capsici* and the control group containing B158-CK and B161-CK mock inoculated with sterile water, three biological replicates were prepared.

The total RNA was extracted from each sample using the RNAprep Pure Plant Kit (TIANGEN, Beijing, China) and then analyzed via 1% agarose gel electrophoresis. The quality of the extracted RNA was evaluated using the NanoPhotometer spectrophotometer (optical density at 260 and 280 nm) (Thermo Fisher Scienific, Shanghai, China). The RNA concentration was determined using the Qubit2.0 fluorometer (Thermo Fisher Scienific, Shanghai, China). The integrity of the RNA was assessed using the Agilent 2100 Bioanalyzer (Agilent Technologies, Beijing, China). The high-quality RNA samples were used for the construction of the cDNA library for the transcriptome sequencing analysis. For each sample, high-quality mRNA was obtained from the total RNA using oligo-(dT) magnetic beads. The mRNA was fragmented (approximately 200 bp) and reverse transcribed to cDNA using random primers. The obtained cDNA fragments were purified and ligated to sequencing adapters. The library was constructed using a PrimeScript 1st Strand cDNA Synthesis Kit (Takara, Dalian, China) and sequenced using the Illumina RNA-seq sequencing platform by Genedenovo Biotechnology Co., Ltd. (Guangzhou, China).

### 4.4. Transcriptomics Analyses

The clean reads generated by filtering the raw data were mapped to the reference *Capsicum* genome (*C. annuum* cv. Criollo de Morelos 334) using the default parameters of HISAT2. New transcripts were assembled using the default parameters of StringTie (https://wiki.gacrc.uga.edu/wiki/StringTie-Sapelo, accessed on 26 October 2020).

Genes that was differentially expressed between samples were identified using EdgeR and DESeq2. The following criteria were used to screen for significant DEGs: false discovery rate ≤ 0.05 and log_2_(fold-change) ≥ 1. The DEGs were subjected to gene ontology (GO) and Kyoto Encyclopedia of Genes and Genomes (KEGG) analyses. Transcription factor genes among the DEGs were annotated according to the Plant Transcription Factor Database.

## 5. Conclusions

In this study, we conducted the screening of resistant varieties B158 and B161 by identifying anthracnose resistance in 17 pepper varieties. The transcription analysis revealed the activation of plant hormone signaling pathways, phenylpropanoid synthesis, and metabolic processes in the defense response of peppers against anthracnose bacteria. Moreover, ARR-B, AP2-EREBP, bHLH, WRKY, and NAC were involved in the defense response triggered by anthracnose bacteria, and especially WRKY and NAC were found to be involved in the defense response mediated by anthrax bacteria, potentially exerting a positive regulatory role. Our research offers a more comprehensive understanding of the resistance of chili red fruits to anthracnose infection.

## Figures and Tables

**Figure 1 plants-13-00527-f001:**
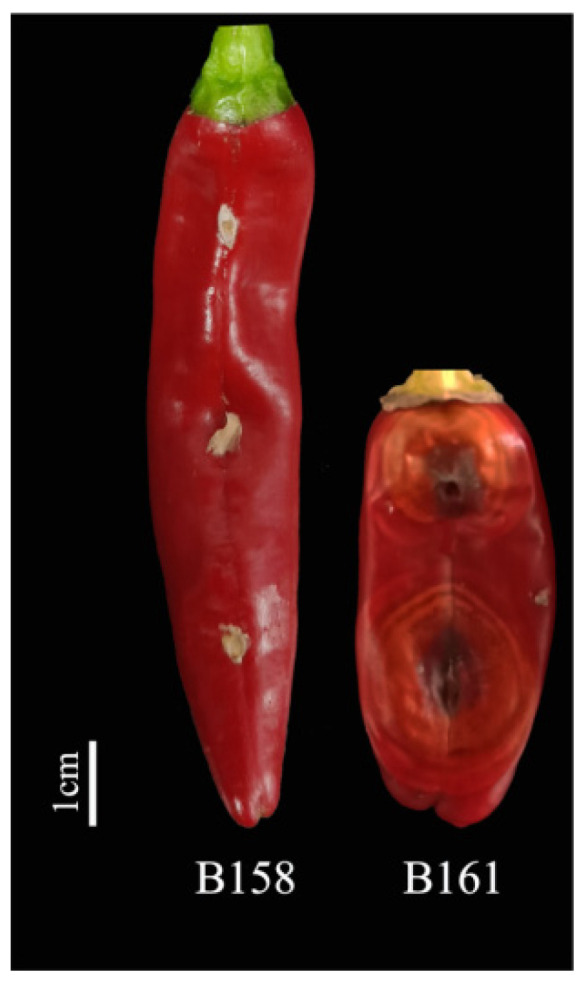
B158 and B161 inoculation anthracnose phenotype.

**Figure 2 plants-13-00527-f002:**
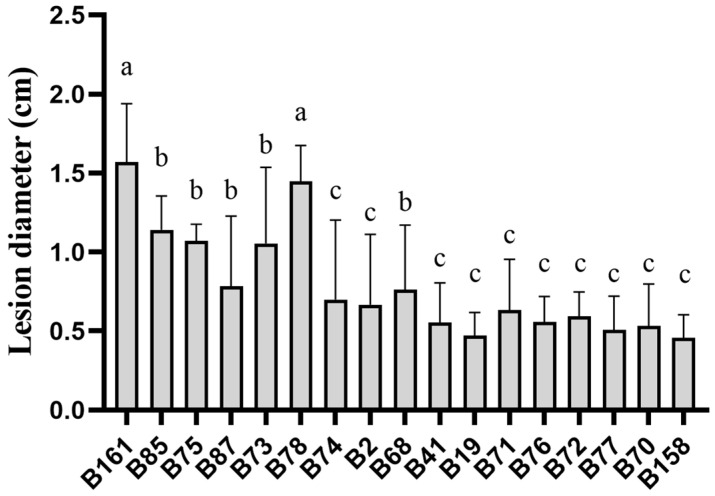
Lesion diameter of pepper varieties. The letters a–c shows significant differences at the *p* < 0.05 level, with statistical significance.

**Figure 3 plants-13-00527-f003:**
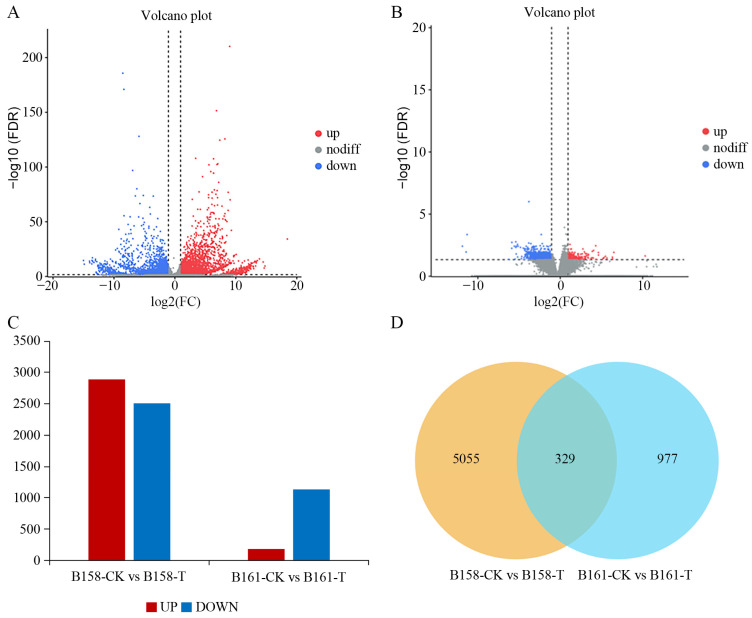
Differential expression analysis. (**A**) Volcanic map of B158-CK vs. B158-T; (**B**) volcanic map of B161-CK vs. B161-T; (**C**) histogram of differentially expressed genes; (**D**) Venn diagram.

**Figure 4 plants-13-00527-f004:**
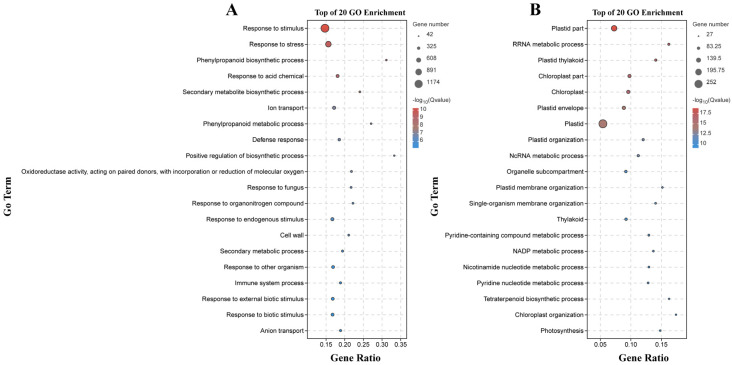
GO enrichment analysis. (**A**) GO enrichment analysis of B158-CK vs. B158-T; (**B**) GO enrichment analysis of B161-CK vs. B161-T.

**Figure 5 plants-13-00527-f005:**
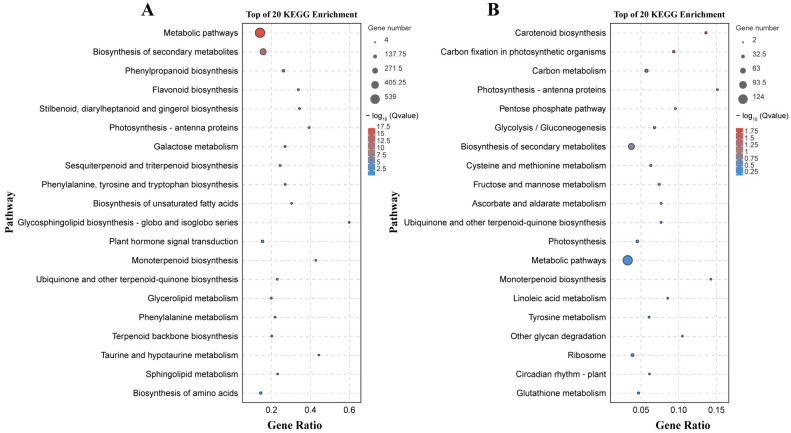
KEGG enrichment analysis. (**A**) KEGG enrichment analysis of B158-CK vs. B158-T; (**B**) KEGG enrichment analysis of B161-CK vs. B161-T.

**Figure 6 plants-13-00527-f006:**
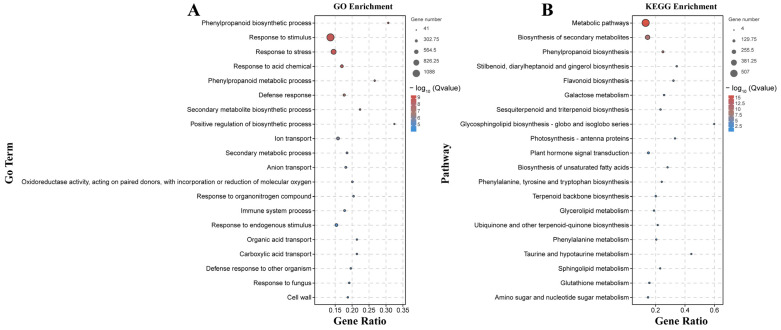
GO and KEGG enrichment analysis of 5055 DEGs. (**A**) GO enrichment; (**B**) KEGG enrichment.

**Figure 7 plants-13-00527-f007:**
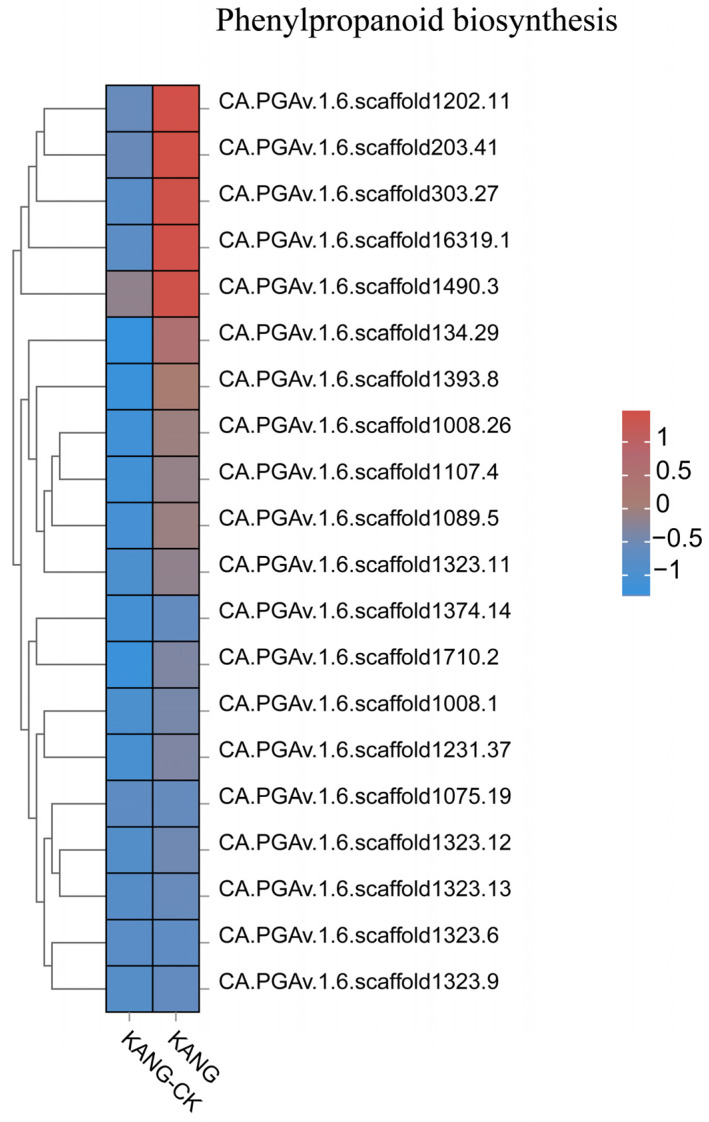
The top 20 gene expression heatmap of phenylpropanoid biosynthesis pathway.

**Figure 8 plants-13-00527-f008:**
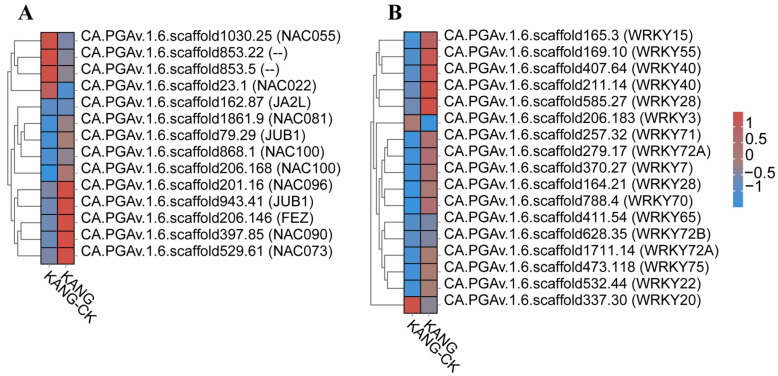
Analysis of transcription factors (TFs). (**A**) Heatmaps of DEGs encoding NACs. (**B**) Heatmaps of DEGs encoding WRKYs.

**Figure 9 plants-13-00527-f009:**
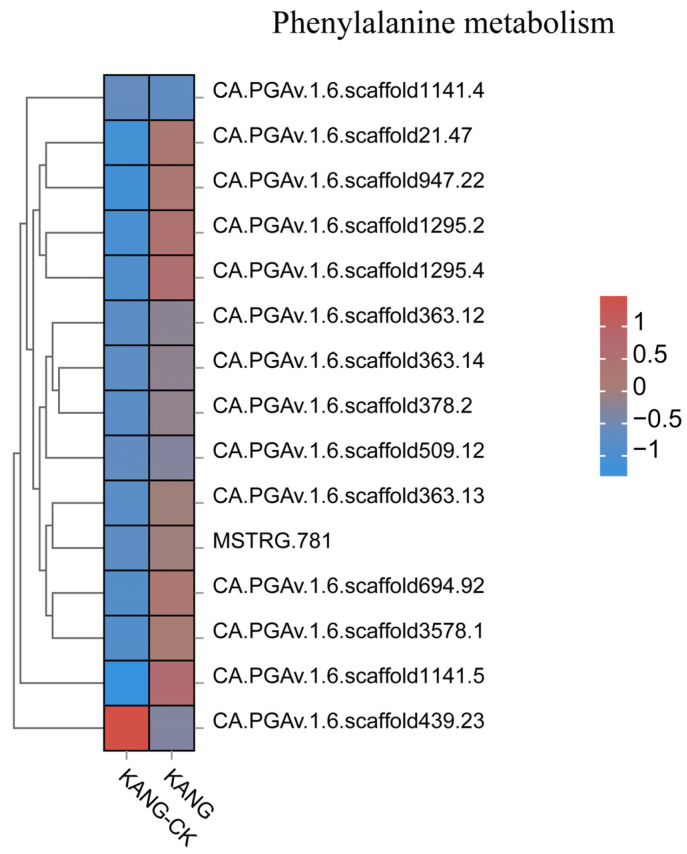
All gene expression heatmap of phenylalanine metabolism pathway.

**Table 1 plants-13-00527-t001:** Overview of the sequencing data.

Sample	Raw Reads	Clean Reads	Q30 (%)	GC (%)
B158-T-1	65,218,986	64,650,424	94.85	43.51
B158-T-2	94,914,578	93,939,304	94.65	44.93
B158-T-3	63,687,242	63,081,338	94.63	43.27
B158-CK-1	65,847,442	65,195,778	94.59	42.28
B158-CK-2	65,128,704	64,532,454	94.66	42.40
B158-CK-3	55,746,766	55,244,818	94.69	42.51
B161-T-1	103,516,244	100,728,566	94.81	45.81
B161-T-2	127,486,314	124,524,228	94.17	49.54
B161-T-3	133,622,868	132,342,810	94.62	49.83
B161-CK-1	92,814,684	91,932,294	94.67	45.18
B161-CK-2	58,120,020	57,515,364	94.56	42.24
B161-CK-3	88,593,188	87,229,950	94.81	43.51

## Data Availability

Data are included within the article and Appendix A.

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
