# Peer review of "Comparative Transcriptomics Analysis Reveals the Differences in Transcription between Resistant and Susceptible Pepper (Capsicum annuum L.) Varieties in Response to Anthracnose"

_plants, 2024, doi:10.3390/plants13040527_

Round 1
Reviewer 1 Report
Comments and Suggestions for Authors
The authors provided evidence on the transcriptomic analysis suggesting pepper resistance to 24 anthracnose may involve the phenylpropanoid biosynthesis and phytohormone signaling path- 25 ways. The article is well-described and properly organized according to the data. Minor corrections are recommended to improve the manuscript.
1. Section 3.2 needs to be improved with the literature on pepper varieties
2. Even the paper describes the Pepper anthracnose severity; why only resistant and susceptible varieties were considered in the protocol?
3. The conclusions need to be separated from the general text as an independent section.
Author Response
- 通过加入转录因子,增强了辣椒品种对生物胁迫响应的描述。但这项研究确定了 5 种对生物胁迫有反应的转录因子,其中大部分与辣椒品种的 WRKY 和 NAC 研究有关。
- 因为抗病感品种的病害指数在17个辣椒品种中最低,最高
- 结论成为一个独立的部分

Reviewer 2 Report
Comments and Suggestions for Authors
Dear Authors,
Thank you for your manuscript submission “Comparative Transcriptomics Analysis Reveals the Differences in Transcription between Resistant and Susceptible Pepper (Capsicum annuum L.) Varieties in Response to Anthracnose.” The manuscript showed the resistant mechanism of pepper to anthracnose disease using comparative transcriptomics analysis. Here are my specific comments:
- Abstract should be rewritten as it does not summarize key points from the study. More quantitative results should be indicated in the abstract
- What is the innovation and significance of this study compared with previous studies? More details are required
- Are there any assumptions for the equation, experimental conditions, sequencing, and data analysis?
- Table 1, how did you define the disease grade as susceptible or tolerant? References are required
- A conclusion section containing key points and results should be provided
- Are there any limitations for the method in this study?
- More details for future research should be provided
Comments on the Quality of English Language
N/A
Author Response
- Abstract rewrite
- Currently, only resistance identification and transcriptome analysis have been conducted, while other validation experiments are still ongoing
- According to the value of the disease index, the resistance and susceptibility were judged, and references were provided
- When conducting resistance identification, it is important to select a representative strain with at least moderate pathogenicity
- We provide a conclusion

Reviewer 3 Report
Comments and Suggestions for Authors
The authors have identified and performed RNA-seq analysis on 2 pepper varieties which were susceptible or resistant to Anthracnose. While informative, the results are very broad and the study lacks focus and originality. A more in-depth analysis could remedy this problem.
Major comments:
The drastic difference in the number of DEGs between B158 and B161 is striking. Is there any information on the genome of these two varieties? The situation to me looks like a key gene is missing/inactive and a defense response is not induced at all in B161.
Related to previous point: apart from WRKY TFs, how do some of the defense markers of SA pathway behave? It would be interesting to check the induction of some of the defense marker genes, such as homologs of PRs from Arabidopsis (These would also include ones that are markers for JA pathways, such as PR4).
All of this is necessary to differentiate between a novel study where the defense mechanisms against Anthracnose are elucidated, vs identifying a hypersusceptible mutant that, by chance, has a key defense gene mutated. For example, just the mutation of a single gene, NPR1, in Arabidopsis renders the plants extremely susceptible and lacking in basal resistance.
Alternatively, if the authors can demonstrate that they identified a novel variant of a key gene that can be bred into susceptible cultivars to enhance their resistance, this will be even more valuable than their current conclusions and strongly increase the significance of their study.
Specific comments
L99: The authors should explicitly state from the very beginning whether in their comparisons if they classify DEGs as upregulated/downregulated in the treated samples compared to mock samples or the other way around (i.e., something like 2887 genes were upregulated in B-158 after the infection)?
L58-60: The authors need to describe which species-pathogen pairs were used, otherwise it seems that they are repeating what has already been done.
LL196-198: What do the authors mean that there are more FLS2 gene upregulated? Are there multiple homologs of FLS2 in pepper or is do they mean the downstream genes? Also, it does not make much sense to focus on FLS2, because it senses bacterial flagellin. The receptor that senses fungal chitin and induces PTI in Arabidopsis is CERK1: were its homolog(s) or the downstream pathway (which overlaps with that of FLS2) more induced in B158 compared to B161? This would provide more relevant evidence for stronger or weaker PTI in different pepper varieties.
Comments on the Quality of English Language
The English is mostly fine. The authors seem to be struggling somewhat in using declarative sentences at times or connecting multiple clauses.
L12: No need for “, which”
L38: change to “value, since it contains capsanthin and capsaicin, which”
LL64 and 65: no need for “that” and “which”
Author Response
- L99: DEGs as upregulated/downregulated in the treated samples compared to mock samples.B158-CK and B161-CK is symbol mock
- L58-60: Describe the species-pathogens Colletotrichum capsici(TJ3-3)
- About FLS2 gene description is delete. Regarding the Arabidopsis receptor CERK1 that you mentioned, teacher, we will focus on further research

Reviewer 4 Report
Comments and Suggestions for Authors
The current transcriptome analysis of susceptible and resistant cultivars of pepper against antracnose disease is very interesting. Howver I am disappointed by the overall analysis and data interpretation od data. The paper requires extensive formatting and renanalysis before critical reviewing. I have highlighted a few major points:
The abstract needs to be properly written. Line 18-19, what does 'susceptible materials' refer to? I would encourage the usage of scientific terms.
Line 26, 'theoretical basis for future investigations', very vague phrase. needs to address how the current study can help in deciphering the resistance/susceptible mechanism of pepper to anthracnose disease.
Line 45: the concept of intrinsic and induced resistance is very controversial. I would suggest the removal of this concept.
For disease scoring shown in table 1, the standard deviation is missing. Also the classification of genotypes based on their disease index is vague. For example in B74 and B78, a difference of 1 can make a variety tolerant or susceptible.
It is strange why only 177 genes are upregulated and vast majority are downregulated in B161 variety. I would suggest a reanalysis of the data. I am unsure why the DEGs in B158 are twice that in B161.
In Line 109, the section on transcription factor looks vague. I would recommend putting it in a separate section and discussing more about The TFs and their possible roles.
Figure 2 and 3 mostly share the same information. I would suggest putting Fig 2 as supplementary fig.
Section 2.5, analysis of DEGs exclusive can be better addressed by identifying the DEGs in resistant pepper cultivar. The section can discuss the resistant genes/families that might contribute to the increased resistance in pepper.
Line 174:'negative regulators of biological processes', what does this mean?
Section 3.2, the discussion of TFs in resistant cultivars needs to be rewritten. It must discuss what type of TFs were found in the study and correlating with other reports how they might contribute to the upregulation of other downstream defense-related targets.
More data on the DEGs related to the plant hormone especially defense hormones and phenylpropanoid pathway that has been highlighted in this study should be shown in the form of heatmaps.
I would suggest incorporating some images showing the differing symptoms in B161 and B158 varieties.
Comments on the Quality of English LanguageThe English language requires extensive editing.
Author Response
- The abstract has been appropriately writen
- Line 18-19, Replaced "susceptible materials" with "susceptible cuitivar"
- Deleted “the concept of intrinsic and induced resistance is very controversial”
-
Provide apepper anthracnose disease grade
- Reanalysis of the data revealed that 1129 genes were up-regulated and 177 genes were down-regulated
- In Line 109,analysis and explanation of transcription factors in a separate section
- Change Figure 2 to an attached image
- Line 174:“negative regulators of biological processes“was simbol up-regulated genes negatively regulate biological processes
- Revise section 3.2 to discuss the importance of transcription factors in response to biological stress,section 3.3 describes how transcription factors regulate downstream genes in response to biological stress
- Add images after inoculation of B158 and B161

Round 2
Reviewer 4 Report
Comments and Suggestions for Authors
I am happy with most of the changes incorporated but emphasize that certain aspects still need to be addressed.
I would prefer a quantitative scale rather than a qualitative one for the disease index. It elaborates more on the disease susceptibility/resistance of the cultivars.
The heat maps or expression visualization pattern of key pathways discussed in the study is key to proving the upregulation of the genes in either susceptible or resistant varieties. Or else it would be difficult for the readers to appreciate and understand the gene profile of the pathways. Hence I would suggest incorporation of expression profiles of respective gene pathways.
Comments on the Quality of English LanguageMinor grammatical errors were detected. Please go through them carefully.
Author Response
1. I would prefer a quantitative scale rather than a qualitative one for the disease index. It elaborates more on the disease susceptibility/resistance of the cultivars.
Response: Thank you for your useful comment. We use quantitative results of lesion diameter instead of disease index (please see page4, Line 72-79, Figure 2).
"By analyzing the lesion diameter of pepper materials, it was found that B158 had the smallest average lesion diameter and B161 had the largest average lesion diameter (Supplementary Table S1). There is a highly significant difference between B158 and B161 at the p<0.01 level. Therefore, B158 is a disease resistant variety and B161 is a disease susceptible variety (Figure 1, Figure 2)".
2. The heat maps or expression visualization pattern of key pathways discussed in the study is key to proving the upregulation of the genes in either susceptible or resistant varieties. Or else it would be difficult for the readers to appreciate and understand the gene profile of the pathways. Hence I would suggest incorporation of expression profiles of respective gene pathways.
Response: Thank you for the useful comment on the manuscript.
We plotted gene expression heatmaps involved in phenylalanine synthesis, plant hormone signaling, phenylalanine metabolism pathways, and transcription factors, please see page9, Figure 7, page10, Figure 8 and page12, Figure 9, Figure S1, S2
3. Minor grammatical errors were detected. Please go through them carefully.
Response: Thank you for the useful comment on the manuscript. Perform grammar and spelling check on the manuscript.
